# Diversity in the internal functional feeding elements of sympatric morphs of Arctic charr (Salvelinus alpinus)

Guðbjörg Ósk Jónsdóttir[1]*, Laura-Marie von Elm[1], Finnur Ingimarsson[2], Samuel Tersigni[1], Sigurður Sveinn Snorrason[1], Arnar Pálsson[1], Sarah Elizabeth Steele[1,3]*

1 Institute of Life- and Environmental Science, University of Iceland, Reykjavik, Iceland, 2 Kópavogur Nature Center, Kópavogur, Iceland, 3 Canadian Museum of Nature, Ottawa, Canada

* goj14@hi.is (GOJ); sar.e.steele@gmail.com (SES)

**Data Availability Statement:** All images of fish and bones are available on repository CharrBonesTVV on Figshare+ (doi:10.25452/figshare.plus.

## Abstract

The diversity of functional feeding anatomy is particularly impressive in fishes and correlates with various interspecific ecological specializations. Intraspecific polymorphism can manifest in divergent feeding morphology and ecology, often along a benthic–pelagic axis. Arctic charr (*Salvelinus alpinus*) is a freshwater salmonid known for morphological variation and sympatric polymorphism and in Lake Þingvallavatn, Iceland, four morphs of charr coexist that differ in preferred prey, behaviour, habitat use, and external feeding morphology. We studied variation in six upper and lower jaw bones in adults of these four morphs using geometric morphometrics and univariate statistics. We tested for allometric differences in bone size and shape among morphs, morph effects on bone size and shape, and divergence along the benthic-pelagic axis. We also examined the degree of integration between bone pairs. We found differences in bone size between pelagic and benthic morphs for two bones (dentary and premaxilla). There was clear bone shape divergence along a benthic–pelagic axis in four bones (dentary, articular-angular, premaxilla and maxilla), as well as allometric shape differences between morphs in the dentary. Notably for the dentary, morph explained more shape variation than bone size. Comparatively, benthic morphs possess a compact and taller dentary, with shorter dentary palate, consistent with visible (but less prominent) differences in external morphology. As these morphs emerged in the last 10,000 years, these results indicate rapid functional evolution of specific feeding structures in arctic charr. This sets the stage for studies of the genetics and development of rapid and parallel craniofacial evolution.

## Introduction

The origins and maintenance of biodiversity are strongly influenced by natural selection acting upon intraspecific variation with fitness consequences [1,2], with many traits directly affecting survivorship and reproduction exhibiting extensive diversification. Adaptive evolution in

25118825). All landmark data is available on Github repository CharrBonesTVV (https://github.com/GudbjorgOskJ/CharrBonesTVV). Flat files with information on specimens are available on Github repository CharrBonesTVV (https://github.com/GudbjorgOskJ/CharrBonesTVV). R-scripts are available on Github repository CharrBonesTVV (https://github.com/GudbjorgOskJ/CharrBonesTVV).

**Funding:** AP: The Institute of Biology at the University of Iceland supported the field expeditions. No grant number. (www.luvs.hi.is). The funder had no role in study design, data collection and analysis, decision to publish, or preparation of the manuscript.

**Competing interests:** The authors have declared that no competing interests exist.

vertebrates entails a substantial degree of variation in feeding behaviour and biomechanics of prey capture and processing that may lead to speciation [3–5]. Morphological adaptations in functional feeding elements are associated with rapid diversification of ecology in many vertebrate groups, sometimes resulting in specialization on diverse prey types within lineages [3,6]. The development of these feeding elements is influenced by an interplay of genes and environmental factors during ontogeny [e.g., 7]. Additionally, developmental remodelling can accommodate ontogenetic niche shifts, for instance within fishes [3,7], and may contribute to intraspecific and interspecific variation in adult morphology.

Extensive complexity and diversification of skull anatomy among fishes correlates with highly diverse feeding ecologies [3,8–10]. Evolution of several functional units [8] has given rise to a diversity of feeding methods (e.g., ram, suction, biting) for optimal foraging on specific prey types, varying widely within and among species [8,9]. Independent evolution of oral jaws, hyoid "jaws" and pharyngeal jaws for prey capture and food processing likely contributed to trophic and morphological diversification [3]. Radiations of ecomorphology and associated prey have occurred widely in fishes, including Pomacentridae [11], Labridae [12], and Cichlidae [13]. While most studies investigate morphological variation among species or higher taxa [8–10,14–16], extensive variation in cranial morphology also exists within species [15,17].

Sympatric polymorphism, where two or more morphs within a species inhabit the same geographic area, has been found in many lacustrine fish species and seems to be promoted by vacant niches, habitat variance, differences in spawning preferences, and relaxation of interspecific competition [18,19]. Resulting morphological divergence is often due to differences in resource use, mainly in benthic vs. pelagic habitats, leading to specializations in traits related to foraging, prey capture and processing [20]. The occurrence of intraspecific morphs that utilize different resources (e.g., prey, habitats) has been widely documented in northern freshwater fishes inhabiting recently de-glaciated systems [20,21]. Salmonid cranial morphology is less derived compared to more divergent groups [17,22], yet within this group many cases of sympatric morphological polymorphism have been described [19,23–25]. Arctic charr (*Salvelinus alpinus*) shows extensive phenotypic variation throughout its geographic distribution [26] and sympatric polymorphism of lake populations is for instance found in Norway [e.g., 27,28], Siberia [e.g., 29], Greenland [e.g., 30] and Iceland [31,32]. In Iceland, the best example of sympatric polymorphism is Lake Þingvallavatn, Iceland's largest natural lake [33]. The lake formed ~10,000 years ago in a rift zone, and is currently a deep lake with lava rocks in most of its banks and shallows [34]. It was colonized by anadromous charr [35,36] that subsequently became isolated. The lake now hosts four morphs of arctic charr (**Fig 1A and 1B**), two benthic: large benthivorous (LB) and small benthivorous (SB) charr and two pelagic: planktivorous (PL) and piscivorous (PI) charr [37,38]. The LB-, SB- and PL-charr are genetically distinct [36,39], while the PI-charr is more heterogeneous genetically, with similarity to either PL- and/or LB-charr [39]. The morphs differ in adult size, diet, habitat use, life history, spawning times, and external and fin morphology [38,40–44]. Two are large (LB and PI) and two small (SB and PL), hereafter referred to as the morph size gradient. Benthic morphs live along the stony littoral bottom and feed mainly on snails [40], and have subterminal mouths (i.e., short lower jaw) and blunt snouts [44,45]. Pelagic morphs more resemble anadromous charr, being more fusiform with a terminal mouth (i.e., long lower jaw) and pointed snout [44,45]. PL-charr feed mostly on crustacean zooplankton, and the larger PI-charr feed mostly on three-spined stickleback [40,45]. Most studies on arctic charr diversity have examined external morphology, except studies of gill raker counts in adults [44], fin ray variation in lab reared juveniles [46] and an unpublished study of internal feeding elements in adults [47]. Notably Ingimarsson [47], using linear measures, found differences among morphs in maxilla and dentary bone shape, relative size of several skull elements, and several teeth traits, with divergence

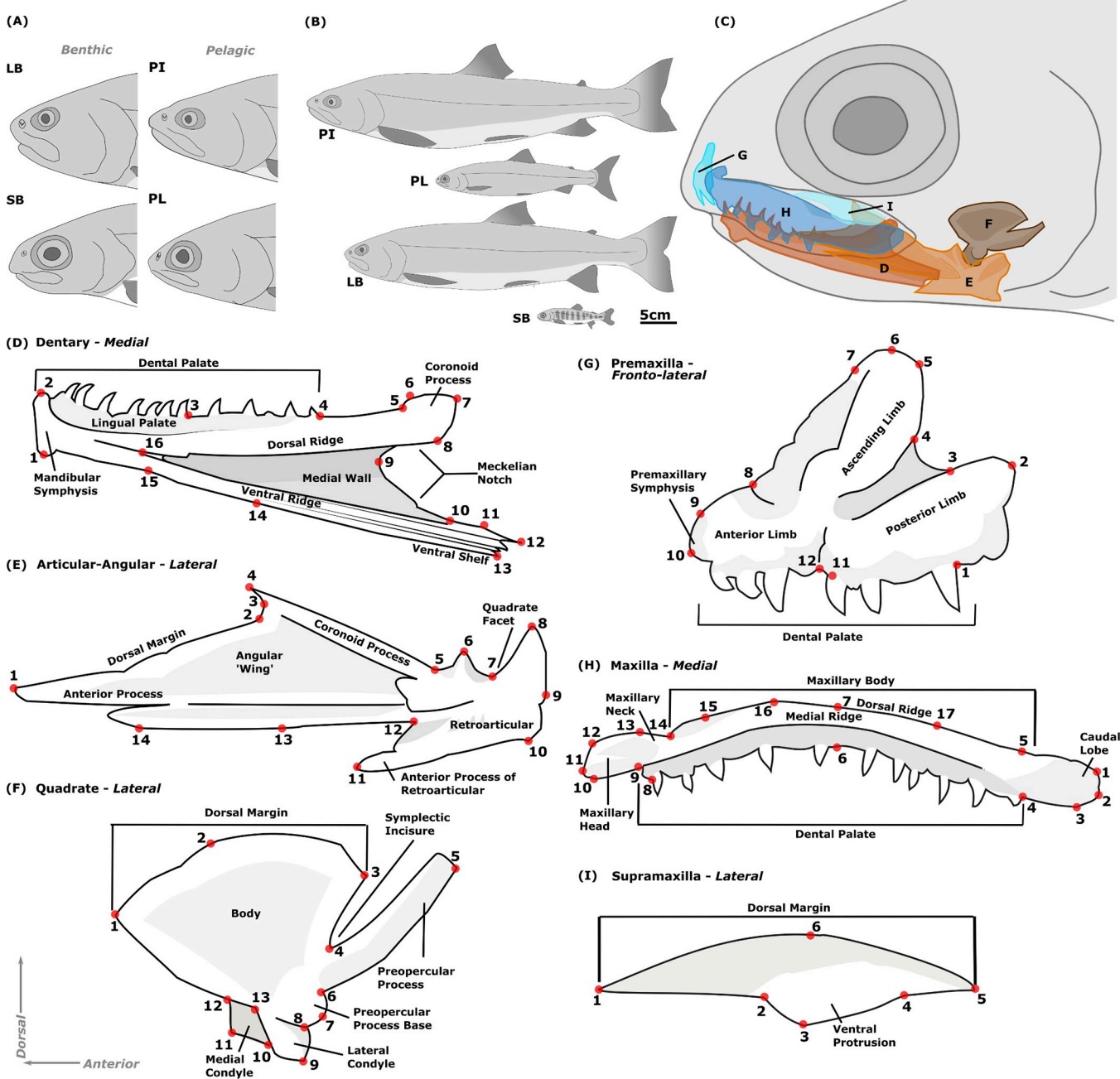

**Fig 1. External and internal morphology of Þingvallavatn arctic charr ecomorphs.** A) Craniofacial variation; B) Postcranial shape and size variation; C) Primary upper and lower jaw bones studied; D–I) Anatomy of the upper and lower jaw bones with landmarks used to capture shape.

mainly along the benthic–pelagic axis. The specific adaptations in cranial morphology by morph and the degree of integration of variation in bone shape are understudied.

To investigate the functional morphology and impact of prey specialization during divergence of arctic charr along the benthic-pelagic axis [32,48], we studied cranial morphology in adults of the four sympatric morphs in Þingvallavatn. To test for divergence and integration in internal feeding structures among morphs, we examined variation in the shape of six cranial bones in the feeding apparatus: dentary, articular-angular, quadrate, premaxilla, maxilla and

supramaxilla (**Fig 1C**). We focus on the following questions and predictions: 1) Is there an allometric component for size or shape of specific bones, and does this vary by morph? We hypothesize that size will influence shape, however, do not expect this allometry to differ by morph. 2) Does the shape of any of those six bones differ among morphs? Given divergence in morph external shape [44], we expect bone shape to differ among the morphs, most likely in the lower jaw and maxilla. 3) Does morphological differentiation occur along a benthic-pelagic axis? Again based on external morphology, where benthic morphs have shorter jaws [44], we expect shortening and compaction of the lower jaw bones (the dentary and/or articular-angular). We also hypothesize proportionally narrower maxilla (main element of upper jaw) in the pelagic morphs, based on observed external morphology [49]. 4) Does integration differ among pairs of bones, between the upper and lower jaw, and across morphs? We expect significant integration between elements, with high integration among bones within the same jaw, but less integration across the upper and lower jaw due to ecological divergence in the lower jaw bones.

## Methods

### Sample and data collection

This study involves sampling and killing of wild fishes, and for such studies a scientific fish fieldwork from the Directorate of Fisheries in Iceland (www.fiskistofa.is) is needed. Arnar Pálsson, an author on this study was in charge and present for all sampling efforts and, for the period of study, had such scientific fish fieldwork permits (#0460/2021-2.0 and #0042/2014-2.13). Sexually mature arctic charr from Lake Þingvallavatn were collected (Fall 2020/21) using composite nets laid overnight (fish died in net) and seine nets (fish were killed by single blow to the head). Fish were randomly sampled, except we aimed for similar representation of morphs and sexes (LB: 49 (21%), SB: 82 (34%), PL: 61 (25%) and PI-charr: 48 (20%), S1 Appendix). Each fish was measured (weight, fork length (FL)), and photographed from left lateral view (Canon EOS 77D, EFS 18–135 mm lens), and sex, sexual maturity, and stomach contents were evaluated by dissection. Morph classification was based on sampling location and date (morphs differ in spawning times and location), external morphology and stomach contents [44,49,50]. Otoliths were extracted from each fish and age determined under 4X magnification in a stereomicroscope. Six jaw bones (dentary, articular-angular, quadrate, premaxilla, maxilla, supramaxilla) were extracted using maceration protocols [50] and photographed (Canon EOS 77D, 50 mm Microlens). The bones from the left side of the head were arranged stereotypically: the dentary and maxilla were photographed and landmarked in medial view, the articular-angular, quadrate and supramaxilla in lateral view, and the premaxilla in fronto-lateral view. To capture external morphology for morph classification, 37 2D landmarks (see **S2 Appendix**) were digitized for each individual using *tpsdig2* (http://www.sbmorphometrics.org/soft-dataacq.html). To study variation in internal feeding structures, we registered 2D landmarks on the six jaw bones with *tpsdig2* (**Fig 1D–1I and S3 Appendix**). All landmarking (external and internal) was conducted by one person (GOJ). To test repeatability of the bone landmarks 168 specimens were landmarked twice. Two landmarks were excluded after the repeatability analysis, LM 12 for premaxilla (**Fig 1G and S3 Appendix**) and LM 15 for the articular-angular (**Fig 1E and S3 Appendix**) because they were added after the first trial. All images of fish and bones are available on Figshare+ (doi:10.25452/figshare.plus.25118825). Landmark data, flatfiles and R-scripts are available on Guithub (https://github.com/GudbjorgOskJ/CharrBonesTVV).

## Statistical analyses

Landmark data were analysed using the R package *geomorph* version 4.0.0 [51,52]. Landmark coordinates were aligned using Generalized Procrustes Analysis (GPA) via the function *gpagen*. Aligned shape variables were then used for all subsequent analyses. Outliers were identified using *plotOutliers* and removed if the bone was damaged. We tested the repeatability of the bone landmark sets in two ways, using Procrustes ANOVA, using *procD.lm*, to characterize measurement error (differences between replicates), as well as two-block partial least squares (PLS) analysis with the function *two.b.pls* to analyze the degree of association between the two replicates (trials). Centroid size (CS) was calculated separately for each of the six bones (CS$_{bone}$; indicates use of CS of a bone, and applied to each bone separately), head (CS$_{head}$) and whole-body (CS$_{body}$). As size is an important predictor for shape in fishes [53,54] and the morphs are known to differ in adult size and proportions [37], absolute size and the relationship between CS$_{body}$ and CS$_{head}$ as well as their relationship with each bone (CS$_{bone}$) were examined. We tested for differences in size and allometry of size measures between morphs using Procrustes ANCOVA using the function *procD.lm*. As salmonid morphology is influenced by sex and age [55,56], we investigated the influence of these variables on morph mean and allometry for size and shape. We then did pairwise tests of groups using the *pairwise* function to test for significant differences between groups. External morphology (whole-body and head shape) was then examined using Procrustes ANOVA to characterize morph differences.

We tested for bone shape differences among morphs and differences in allometric relationships of bone shape and bone size across morphs using Procrustes ANCOVA, using *procD.lm*, with CS$_{bone}$, morph and their interaction as independent variables. Null (shape = CS$_{bone}$), reduced (shape = CS$_{bone}$ + morph) and full models (shape = CS$_{bone}$ x morph) were compared to determine best fit. When the full model had the best fit, differences in allometric trajectories were investigated using *pairwise* with CS$_{bone}$ as a covariate. We tested for significant differences in both vector angles (vector correlation) and vector lengths (rate of shape change per unit covariate), considering significance among both tests as robust support for biological differences. For all tests α = 0.008 was chosen, using the Bonferroni method as a conservative correction for multiple comparisons (six bone datasets). Additionally, we considered comparisons with Z scores <3 as poorly supported, and interpreted results below that as not biologically meaningful and chose the reduced model for further interpretation. Differences in allometry among morphs were investigated using *plotAllometry* and visualized using predicted values from regression analyses. Principal Component Analysis (PCA, using *plotTangentSpace*) was used to visually represent shape variation among individuals and morphs. For bones that did not show significant differences in allometry among morphs, residuals from a regression of shape on size (size-corrected shape variables) via *procD.lm* were used for visualization. Shape variation along PC axes was summarized with deformation grids and vector plots from *shape.predictor*.

To analyze integration among bones, pairs of bones were compared using two-block partial least squares (PLS) analysis with the function *two.b.pls*. We compared elements within and among bones of the upper and lower jaws. The effect sizes were compared to determine if there was significant variation in levels of integration across different bone pairs or morphs using *compare.pls*. For all analyses we used randomized residuals in 1000 permutation procedures to assess statistical significance and estimate the effect sizes (*RRPP* implemented in *geomorph*, [57]). For integration tests α = 0.003 was chosen as a conservative threshold (Bonferroni correction for 15 pairwise comparisons among bones). Figures were generated using R and formatted using Inkscape.

## Results

### Differences in body size, sex and age of sympatric morphs

To capture the morphological diversity in the four sympatric morphs from Lake Þingvallavatn we studied 240 sexually mature individuals. As was expected, morphs differed in size (fork length FL, F = 350.94, p << 0.001), log weight (F = 381.28, p << 0.001) and age (F = 56.75, p << 0.001, S4–S6 Appendices). SB- and PL-charr were smaller at sexual maturity than the LB- and PI-charr. The weight was influenced by sex (p < 0.001), and there was a significant sex by morph interaction (p = 0.009). The external whole-body (p = 0.001) and head (p = 0.001) shape differed by morph, with pelagic morphs having a more fusiform body shape and a more pointed snout (S7 Appendix). There was significant sexual dimorphism in whole-body shape ($R^2$ = 0.04, p = 0.001, S8 Appendix), with males having a more humped back compared to females (not a craniofacial trait). These results corroborate morphological descriptions of the morphs [44] and known sexual dimorphism [56]. There was weak sexual dimorphism ($R^2$ = 0.01, p = 0.007) for shape of head and all six bones ($R^2 \leq$ 0.02, S8 Appendix) that will not be discussed further. Age was strongly correlated with FL ($R^2$ = 0.7), but always explained a lower proportion of variation than FL (for external shape, bone size and shape variation). Therefore, FL was used as a covariate in allometry analyses of bone size relative to body size.

### Allometry of bone size and shape

Examination showed that all six bone landmarks sets were highly repeatable (**S9, S10** and **S11 Appendix**), the association between replicates was high (r-PLS ~ 0.84–0.99) and variation due to measurement error was low relative to variation among individuals ($R^2$ = 0.002–0.005, vs. $R^2$ = 0.89–0.94, **S11 Appendix**). Thus, the landmarks developed here for shape of these six bones in salmonids appear to be robust.

Predictably, the size of all bones ($CS_{bone}$) increased with body length ($\log_e$ FL, $R^2 \geq$ 0.94, p = 0.001, **Fig 2** and **S6** and **S12 Appendices**). Morph and morph by size interaction, while significant, accounted for a small fraction of the variation ($R^2 \leq$ 0.04, S6 Appendix). Additional testing revealed small differences in bone size allometry between pairs of morphs (**S13** and **S14** Appendices). These differences mostly reflected differences between larger (PI-charr) and smaller (SB-, PL-charr) morphs. PI-charr have significantly larger dentary bones than all other morphs (p = 0.001), as well as significantly larger articular-angular, maxilla, and supramaxilla bones than both LB- and PL-charr (p < 0.002). PL-charr also have a significantly smaller quadrate, premaxilla and supramaxilla compared to SB-charr (p = 0.001). As $CS_{bone}$ was highly correlated with body size ($CS_{body}$ and $\log_e$ FL) and a common allometric relationship was found among morphs, we used $CS_{bone}$ as size measures in subsequent analyses.

For all bones, shape was significantly affected by size ($CS_{bone}$), the allometric relationship explained 20–30% of shape variation for dentary, articular-angular, quadrate, and maxilla ($R^2$ = 0.2–0.3, **Table 1**, for premaxilla and supramaxilla, bone size only explained 10–15%). Note, for the dentary and articular-angular morph explained a similar amount of the shape variation as size (**Table 1** and **S15 Appendix**). The allometry of all bones varied by morph (p = 0.001, **Table 1**), except for the quadrate, however the effects were small for maxilla and supramaxilla. Further comparisons of vector correlation angles and lengths indicated that allometric relationships vary only markedly by morph in the dentary (**S16–S18** Appendices). For this bone, while the common allometry and morph effects on shape were much larger than the interaction, there was significant support for allometric differences among several morph pairs (**S16 Appendix**). Specifically, the PI-charr differed significantly in allometric trajectory orientation from all other morphs (p $\geq$ 0.003), converging on a similar shape as LB-charr at larger size.

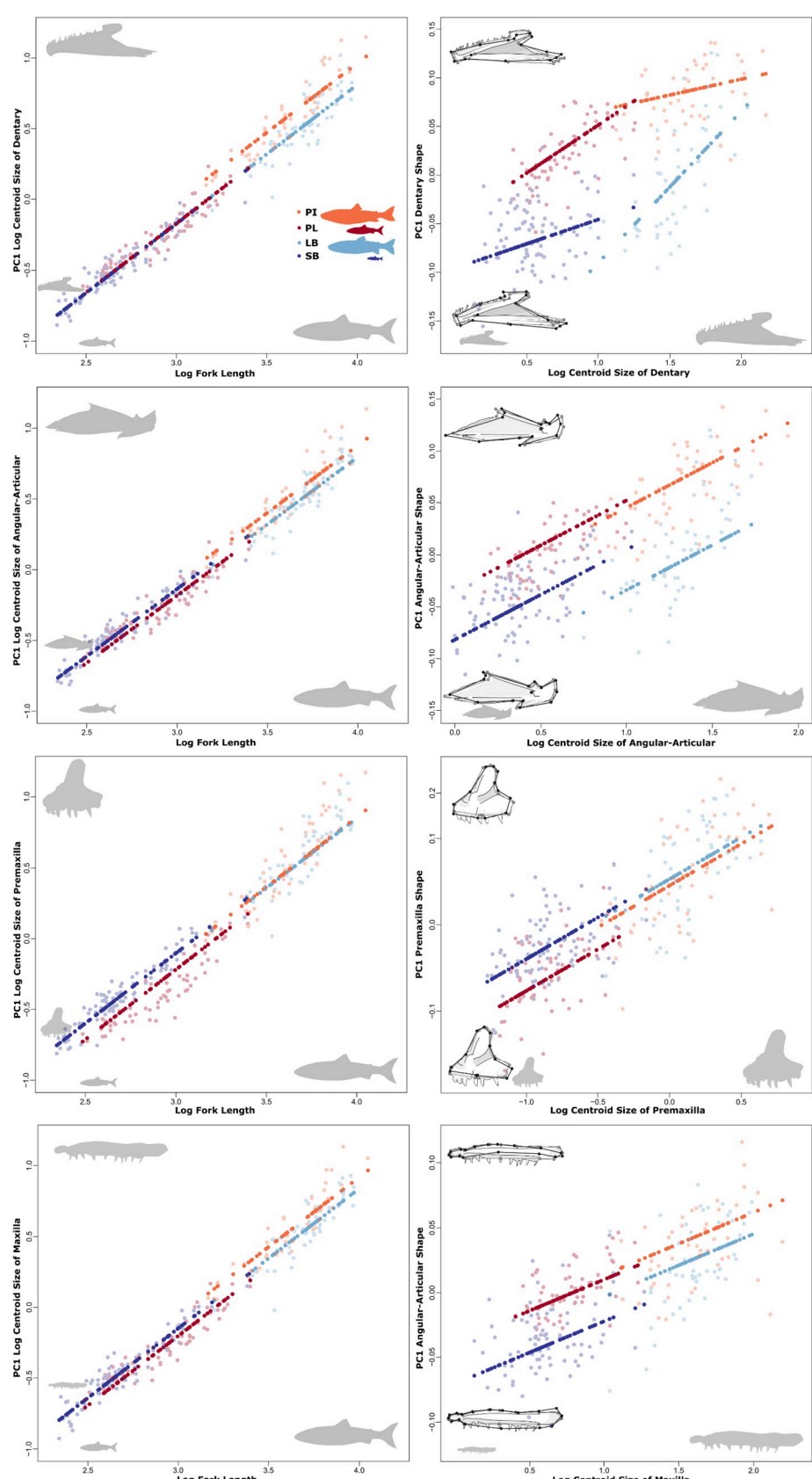

**Fig 2. Size and shape allometry of dentary, articular-angular, premaxilla and maxilla (from top to bottom, inset pictures of each bone).** Left panels: relationships between bone size (y) and fork length (x) by morph; Right panels: relationships between bone shape (y) and bone size (x). Inset are the associated shape changes related to each component, grey outlines the mean shape, and black the two extremes for each PC. Shown are values for individuals (open circles) and the predicted values (filled) of regressions for both bone size ($CS_{bone}$) vs body size ($\log_e$ FL) or bone-shape versus bone-size. $\log_e$, natural log transformation.

While the allometric trajectories of LB- and SB-charr had similar lengths, the rate of shape change was higher in LB- compared to SB-charr (p = 0.001). Therefore, we present below the mean shape differences among morphs from the full model ($CS_{bone}$ X morph) for the dentary but for all other bones results from the reduced model ($CS_{bone}$ + morph), as both common allometry and morph affected shape variation significantly (p = 0.001). All visualizations, unless otherwise noted, have been corrected for size to remove the effects of allometry on shape. We focus first on the lower jaw bones and then the upper jaw.

## Benthic and pelagic divergence in shape of specific lower jaw bones

For the dentary and articular-angular, the morph effect on shape variation was similar in magnitude as the common allometric effect (20–30% of shape variation, **Table 1**). For the other bones the size (allometry) was considerably stronger.

Most dramatically, pairwise tests revealed differences in dentary shape between all morph pairs across the benthic-pelagic axis, except among the two smaller morphs (SB- and PL-charr, p = 0.119, **Table 2**). Also, the comparison PI-SB-charr was weakly supported (Z < 3). No shape differences were within morphotype (LB-SB-charr, PL-PI-charr). The dentary shape differences by morph were visualized using PCA (Fig 3), and due to variation in allometry among morphs, size effects were not removed. The morphs separated along a benthic-pelagic axis on PC1 (50% of the total variation) with pelagic morphs, on average, having an elongated dentary, increased length of the dental palate, more acute angle between ridges, and vertically inclined lingual palate relative to benthic morphs, which aligns with the hypothesis of shorter lower jaw in benthic charr. PC1 represents elongation of the dentary bone, with positive values representing a more elongate shape, with extension of the dental palate containing the teeth, and negative values representing a compact shape with reduction in the length of the dental palate. Positive values were also associated with a more acute angle between the dorsal and central ridges, reducing the size of the Meckelian notch, as well as a more vertical inclination of the lingual palate. PC2 explained 13% of the total variation, and mainly split the morphs by size, the larger morphs (LB-, PI-charr) having positive values and taller bones. The associated shape

**Table 1. The influence of size (bone centroid size) and morph on shape variation in 6 jaw bones.** Analysed with Procrustes ANCOVA, testing for main and interaction terms. For all bones, except quadrate, was the full model with interaction the best fit.

| Bone Shape | $\log_e$ CS Effect | | | | Morph Effect | | | | Morph x $\log_e$ CS Interaction Effect | | | |
|---|---|---|---|---|---|---|---|---|---|---|---|---|
| | $R^2$ | F | Z | P | $R^2$ | F | Z | P | $R^2$ | F | Z | P |
| Dentary | 0.237 | 107.58 | 5.29 | **0.001** | 0.228 | 34.49 | 6.35 | **0.001** | 0.026 | 3.89 | 5.09 | **0.001***|
| Articular-angular | 0.190 | 72.95 | 5.26 | **0.001** | 0.184 | 23.43 | 7.43 | **0.001** | 0.020 | 2.61 | 3.56 | **0.002** |
| Quadrate | 0.298 | 102.95 | 6.32 | **0.001** | 0.020 | 2.35 | 2.98 | **0.002** | - | - | - | - |
| Premaxilla | 0.165 | 55.15 | 9.14 | **0.001** | 0.110 | 12.27 | 9.53 | **0.001** | 0.030 | 3.33 | 5.19 | **0.001** |
| Maxilla | 0.199 | 67.56 | 7.75 | **0.001** | 0.096 | 10.87 | 7.54 | **0.001** | 0.022 | 2.53 | 2.83 | **0.001** |
| Supramaxilla | 0.099 | 27.66 | 6.27 | **0.001** | 0.074 | 6.90 | 5.29 | **0.001** | 0.027 | 2.55 | 2.75 | **0.003** |

*Pairwise tests (S17 and S18 Appendices) indicated only the dentary had true differences in allometry by morph.

$R^2$ = coefficient of determination; F = F-statistic; Z = Z-statistic; P = P-value, ≤ 0.008 in bold (significant after Bonferroni correction).

**Table 2. Morph differences in bone shape estimated by pairwise distances in mean shape, from the best fit ANCOVA models for each bone (Table 1).**

| Bones | Morph pairs | d | UCL (95%) | Z | P-value |
|---|---|---|---|---|---|
| Dentary | LB-PI | 0.176 | 0.129 | 3.91 | **0.001** |
| | SB-PL | 0.106 | 0.110 | 1.24 | 0.119 |
| | LB-PL | 0.154 | 0.131 | 3.15 | **0.001** |
| | PI-SB | 0.131 | 0.117 | 2.65 | **0.005** |
| | LB-SB | 0.070 | 0.072 | 1.47 | 0.070 |
| | PI-PL | 0.042 | 0.051 | 0.60 | 0.270 |
| Articular-angular | LB-PI | 0.088 | 0.023 | 6.77 | **0.001** |
| | SB-PL | 0.063 | 0.022 | 4.43 | **0.001** |
| | LB-PL | 0.104 | 0.035 | 5.12 | **0.001** |
| | PI-SB | 0.059 | 0.038 | 3.04 | **0.001** |
| | LB-SB | 0.055 | 0.039 | 2.72 | **0.005** |
| | PI-PL | 0.038 | 0.035 | 1.98 | 0.021 |
| Quadrate | LB-PI | 0.024 | 0.025 | 1.54 | 0.065 |
| | SB-PL | 0.022 | 0.021 | 1.73 | 0.045 |
| | LB-PL | 0.037 | 0.037 | 1.58 | 0.050 |
| | PI-SB | 0.048 | 0.038 | 2.65 | **0.006** |
| | LB-SB | 0.038 | 0.040 | 1.43 | 0.080 |
| | PI-PL | 0.046 | 0.035 | 2.91 | **0.004** |
| Premaxilla | LB-PI | 0.094 | 0.036 | 5.62 | **0.001** |
| | SB-PL | 0.087 | 0.030 | 6.30 | **0.001** |
| | LB-PL | 0.086 | 0.054 | 3.70 | **0.001** |
| | PI-SB | 0.098 | 0.053 | 4.56 | **0.001** |
| | LB-SB | 0.034 | 0.054 | -0.38 | 0.653 |
| | PI-PL | 0.049 | 0.053 | 1.31 | 0.090 |
| Maxilla | LB-PI | 0.024 | 0.019 | 2.35 | 0.010 |
| | SB-PL | 0.051 | 0.017 | 5.93 | **0.001** |
| | LB-PL | 0.053 | 0.028 | 4.06 | **0.001** |
| | PI-SB | 0.048 | 0.031 | 3.28 | **0.001** |
| | LB-SB | 0.033 | 0.032 | 1.83 | 0.037 |
| | PI-PL | 0.050 | 0.028 | 4.04 | **0.001** |
| Supramaxilla | LB-PI | 0.022 | 0.038 | 0.002 | 0.490 |
| | SB-PL | 0.083 | 0.033 | 4.60 | **0.001** |
| | LB-PL | 0.061 | 0.056 | 1.89 | 0.028 |
| | PI-SB | 0.102 | 0.061 | 3.41 | **0.001** |
| | LB-SB | 0.096 | 0.059 | 3.37 | **0.001** |
| | PI-PL | 0.051 | 0.056 | 1.33 | 0.092 |

UCL = upper confidence level; Z = Z-statistic; P-value $\leq$ 0.008 in bold (Bonferroni correction).

changes mainly affected the height of the whole bone, with negative PC values associated with thinner bones overall, reduced height of the mandibular symphysis, coronoid process, and ventral shelf. Together, PC1 and PC2 separated all morphs clearly in shape space, but they overlap (possibly due to overlapping sizes ranges and/or more size variation within LB-charr).

Pairwise tests for articular-angular shape also revealed differences between all morph pairs across the benthic-pelagic axis. Biologically meaningful differences were not found within morphotypes (Z<3, see **Table 2**). The morphs aligned along a benthic-pelagic axis on PC1 which explained 37% of the variation. The two pelagic morphs were largely overlapping

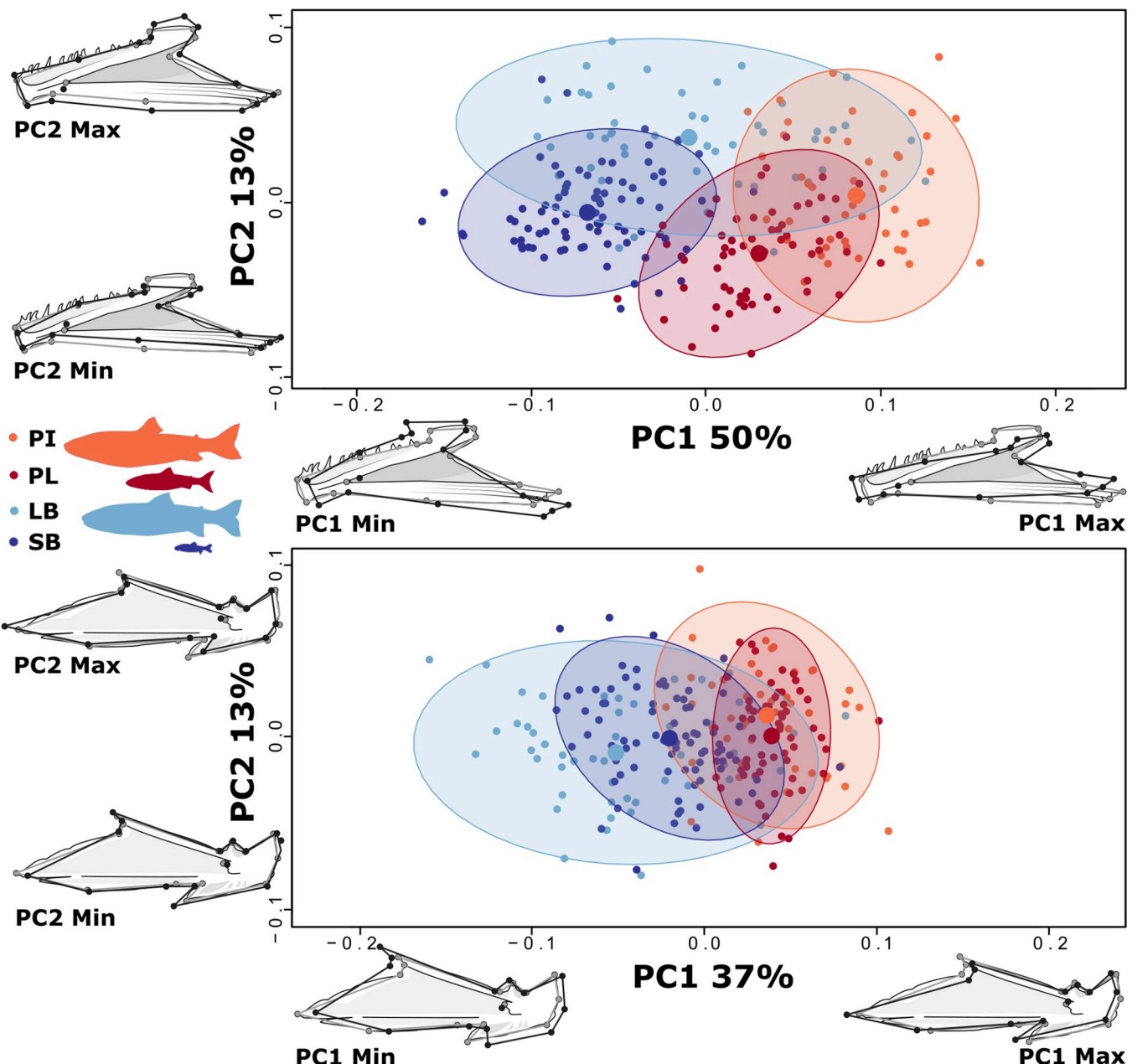

**Fig 3.** Strong morph effects on shape variation in the lower jaw bones, dentary (top) and articular angular (bottom). For the dentary, size effects were not removed, but effects of size were removed for articular angular (and other bones, see below). PC-plots of specimens and deformation grids show how shape differences by morphs (grey outlines the mean shape and black lines the two extremes for each PC). None of the shape warps are magnified. Each dot represents an individual and the ellipses 95% CI for the distribution by morph (large dot represents the mean for each morph in these dimensions). PC1 and 2 explain 50% and 13% of the variation in dentary, respectively. For the articular angular (bottom) PC1 and 2 explain 37% and 13%.

(positive values), with elongated anterior process of the angular "wing", shorter coronoid process, and narrower quadrate facet. The benthic morphs had more shape variation (especially the LB-charr, as in dentary) and more compact bone shape and wider quadrate facet overall, with LB-charr individuals being most compact (**Fig 3**). PC1 represents changes in relative sizes of the anterior process of the angular 'wing', retroarticular, and quadrate facet. Individuals with positive PC1 values show anterior extension of the anterior process of the angular "wing",

increase in the size of the anterior process of the retroarticular but dorso-ventral compression of the retroarticular, and narrow quadrate facet. Negative values associate with a more compact shape overall, particularly with restrictions to the length of the anterior process of the angular "wing", however the coronoid process is extended anteriorly, a more square retroarticular, and wider quadrate facet. PC2 explains 13% of the variation and mainly represents differences in the ventral process of the retroarticular, with individuals with negative PC values with a shorter process. The morphs did not separate along PC2.

In sum, the morph differences in the articular-angular and in particular the dentary were large. In contrast, for the third lower jaw bone the quadrate morph had weak effects on shape ($R^2$ = 0.02, Z < 3, Table 1). PCA and pairwise tests for mean quadrate shape differences did not reveal shape divergence among morphs (**Table 2** and **S19 Appendix**).

## Weaker divergence in shape of upper jaw bones

For the upper jaw bones the magnitude of the morph and size effects on shape were similar (~10–15%, Table 1) but notably lower than in the dentary and articular-angular. Morph had strongest effect on the premaxilla and weakest (nearly none) on the supramaxilla. The analyses were done on size corrected data, assuming common allometry.

Pairwise tests for mean premaxilla shape differences among morphs indicated significant differences in all morph pairs across the benthic-pelagic axis (p ≤ 0.001, **Table 2**). No differences were found within morphotype. The morphs separated along a benthic-pelagic axis on PC1 (25% of the variation) which captured morphological shape differences of the posterior and anterior limbs. The PL- and PI-charr had positive PC1 values with elongated and narrow limbs, while SB- and LB-charr overlapped with more negative PC1 values, having compact and wide limbs (**Fig 4**). PC2 explained 14% of the variation, representing variation due to differences in the curvature of ascending limb but did not separate morphs.

For maxilla shape, pairwise tests of mean shape between morphs revealed significant separation between all morphs (p = 0.001), except the SB-LB-charr and LB-PI-charr (**Table 2**). Therefore, maxilla diverged neither exclusively along the benthic-pelagic axis nor the size gradient. The morphs separated on the benthic-pelagic axis on PC3 (10% of variation). This component represents the height of the maxilla (positive: tall and negative: short) aligning with our hypothesis, but the distributions overlapped substantially (**Fig 4**). PC1 explained 33% of the total variation, but it represents technical variation (either due to rotation or tilting of the bone during photographing, that was randomly distributed with respect to morph). PC2 (23% of total variation), representing biological variation due to differences in the curvature of the maxillary head and angle of the maxillary neck, and separated the PL slightly from other morphs. In combination, PC2 and PC3 indicate that PL-charr has reduced morphological variation and appears to have a thinner maxillary body and caudal lobe than the other morphs.

Pairwise comparisons of mean supramaxilla shape differences among morphs indicated SB deviated from all other morphs (p = 0.001, Table 2). PC1 and PC2 explained 62% of variation, but no combination of PCs revealed clearly how SB-charr shape deviated (S19 Appendix). We note the morphological variation in SB appeared reduced relative to other morphs.

## Covariation in shape change among bones

Lastly, we tested for covariation in shape among bones. Within the upper jaw and within the lower jaw, shape covaried significantly among all bones (p = 0.001). The degree of covariation was greatest for the dentary and articular-angular (r-PLS = 0.92; Z = 7.08, S20 Appendix) and much higher compared to the other pairs (r-PLS = 0.54–0.69). Among the jaws, all covariation among bones was significant (p = 0.001) and the degree of covariation was moderate (r-PLS ~

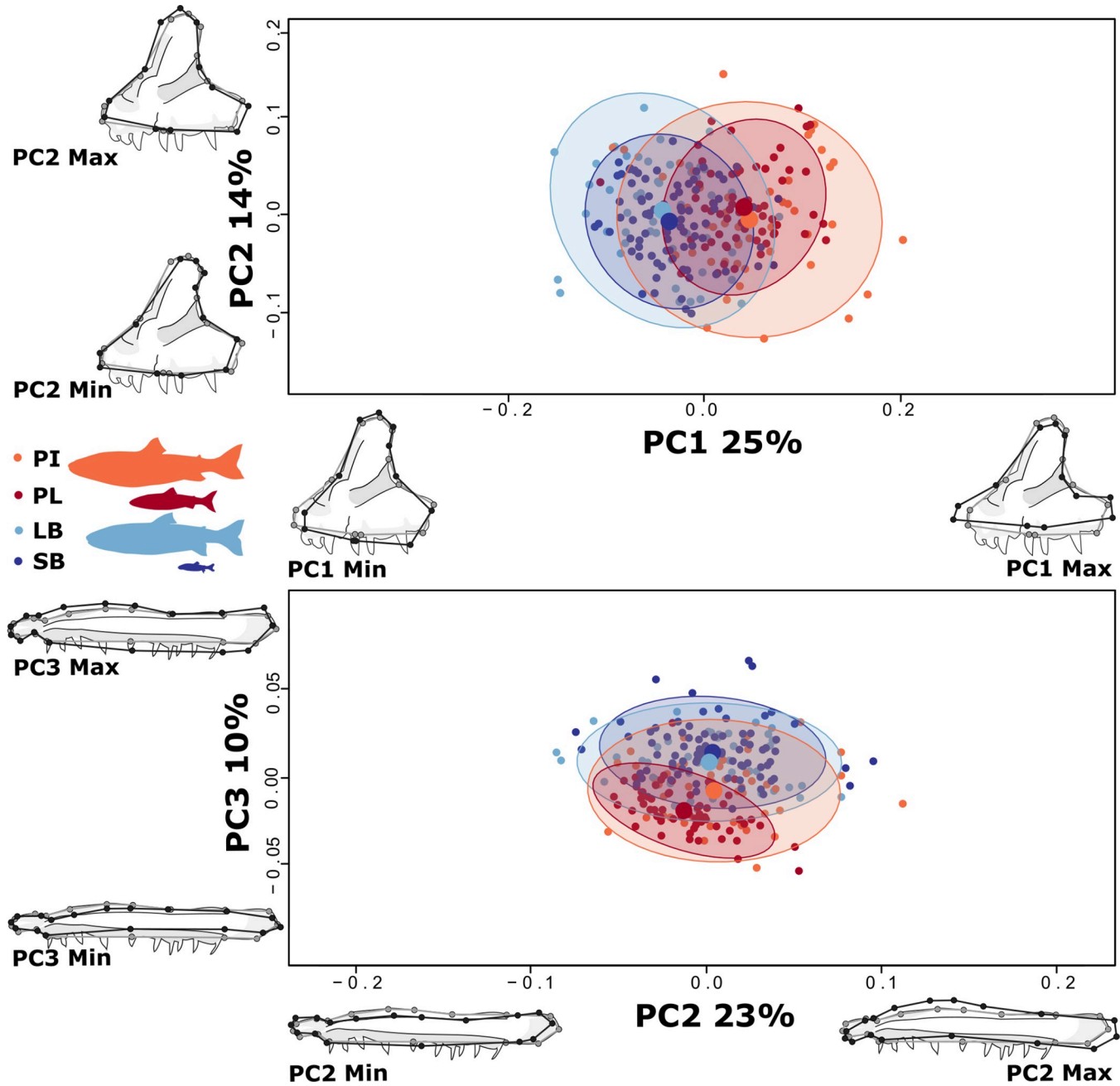

**Fig 4. Pronounced divergence in premaxilla of the upper jaw and subtle changes in maxilla shape.** Size corrected PC-plots of specimens and deformation grids showing how the morphs differ in mean (top) premaxilla and (bottom) maxilla shape (grey outlines the mean shape and black lines the extremes for each PC). None of the warps (are magnified, representation of PC plots and warps as in Fig 3. For premaxilla (top) PC1 and PC2 explain 25% and 14% of the variation respectively. For maxilla (bottom) PC2 and PC3 explain 23% and 10% respectively (PC1 was biased by sampling error, and not depicted).

0.54–0.78, S20 Appendix). We did not find significant differences in effect size among pairs in the lower jaw, however in the upper jaw integration between the premaxilla and supramaxilla was found to be lower (p = 0.006) than all other comparisons, and integration between the maxilla and supramaxilla was found to be higher (p = 0.002) (S20 Appendix). In addition, analyses did not reveal significant differences in effect size between morphs for any of the bone

pairs (comparisons limited to bones that showed significant covariation or biological importance as described above, quadrate and supramaxilla excluded).

## Discussion

Evolutionary divergence along the benthic-pelagic axis occurs repeatedly among fishes [58–61]. Body elongation is a consistent trend along the benthic-pelagic axis in marine and freshwater fishes [58] as well as fin adaptations for sustained swimming in pelagic environments and manoeuvrability in benthic habitat [59,62]. Craniofacial responses to benthic and pelagic prey resources are quite complex [3,63–65]. Prey specialization along this axis leads to morphological adaptation of multiple traits that can be either integrated or modular, with known examples of functional convergence via a range of morphological solutions [66]. Salmonids are known for extensive variation in feeding specializations and morphology but to date few studies have investigated variation in internal structures among or within a salmonid species [except see, 17,23,24,46,47,67,68]. Notably, in addition to pioneering work of Ingimarsson [47], one study reported differences in dentary and articular-angular shape in parr, between several salmonid species and also within rainbow trout (*Oncorhynchus mykiss*, [17]).

We report morph differences in body, head and bone shape, with bones showing more pronounced differences. The whole-body and head shape differences by morph are consistent with previous studies [44,56]. We examined six cranial bones tied directly to feeding and prey capture in recently evolved sympatric morphs of arctic charr, and the results corroborate our hypothesis (1), of clear allometric shape variation in all six bones. Surprisingly, shape allometry did differ between morphs for the dentary. In accordance with hypothesis (2) we found the strongest benthic-pelagic contrast in the most anterior bones in lower jaw bone (dentary). We did not expect premaxilla divergence, however. These bones interact directly with prey, but shape differences were also seen in the articular-angular and maxilla. Shape differences in dentary and maxilla are consistent with hypothesis (3) (shorter lower jaw and more round maxilla in benthics). Also, curiously, for both the dentary and articular-angular the morph effects (benthic-pelagic axis, on PC1 for both) were stronger than the effects of size (PC2 for both). In contrast, the shape of the two posterior bones, quadrate and supramaxilla, did not vary strongly between morphs. Following hypothesis (4), shape variation in bones were highly correlated with each other, with bones of the same jaw more integrated than bones among jaws. We discuss these results in the context of functional feeding morphology and its mechanics, the putative genetic and environmental sources of these traits and suggest future studies of craniofacial diversity in salmonids. Here we focus on cranial bone shape and potential relationships to prey and mechanics of prey capture, but acknowledge that postcranial divergence allows access to preferred prey habitat and also aids in locating, approaching, and capturing individual prey [3,59].

### Bone shape variation and functional divergence

We found strong divergence in the lower jaw along the benthic-pelagic axis. The characteristic subterminal mouth of the benthic morphs associates with a shortening of the dentary and to lesser extent articular-angular (relative to the quadrate facet) aligns with our hypothesis (3). Additionally, the anterior region of the dentary around the mandibular symphysis is angled ventrally relative to pelagic charr (**Fig 3**). This may aid in creating a more optimal angle to grip attached prey or perform scraping behaviours on a surface to dislodge prey, such as snails. The benthic morphs may use a combination of biting and ramming behaviours, as suction forces are likely weak. Taller dentary and articular-angular bones could be adaptations for feeding on benthic prey, increasing size of the mouth cavity and efficiency of consuming benthic

organisms [69]. The coronoid processes of the dentary and articular-angular are enlarged in benthic charr, which could associate with a larger mandibular portion of the adductor mandibulae and a larger aponeurosis maxillaris [70], thus allowing for greater bite forces in benthic morphs which could aid in dislodging attached prey items [63,71,72]. Relatively more elongated lower jaw was expected and observed in pelagic morphs. Consistently, previous study on lower jaw bone shape in juveniles of *O. mykiss* found the anadromous population (with presumed pelagic lifestyle) had a more elongated and streamlined lower jaw than the resident population [17]. The authors postulated that elongated, streamlined lower jaws could be adaptations to the pelagic ocean environment. It would be interesting to see if this holds in more salmonid populations, and also if this is accompanied by notable allometric changes in the dentary as we found here.

In the upper jaw, we found premaxilla shape to diverge along the benthic-pelagic axis more strongly than the maxilla. PL-charr had smaller premaxilla overall (importantly relative to SB which is of same size), with shorter ascending limbs and narrow anterior and posterior limbs. Reductions in the height of the ascending process and elongation of the anterior-posterior axis of the premaxilla has been found in other pelagic fishes [64,73,74] that use ram feeding in the water column [75] while suction feeders with highly protrusible jaws have long ascending limbs [76]. Moreover, shortening and steeper angles of the ascending limb of the premaxilla can also be associated with increased biting forces [77] and possibly linked to biting behaviours. The ascending limb of the premaxilla has been found to extend more dorsally in pelagic fish [78] and in highly protrusible jaws [79], however, in our study the ascending limb was proportionally longer in the benthic morphs. As the premaxilla, and the skull as a whole, is relatively rigid in salmonids [sensu 70], the biomechanics of similar morphological changes in fishes with protrusible jaws could be incomparable. For the maxilla, the third shape PC associated with the benthic-pelagic axis. The maxilla of fishes is generally involved in the protrusion of the upper jaw maxillary apparatus, while moveable in salmonids such function may not be present [70] and the proposed movement of these components to aid in salmonid feeding [80] has been criticized [70]. The shape change seen in the bone matches hypothesis 2, based on observations done on external photographs in pelagic charr [49].

Neither the quadrate nor supramaxilla diverged along a benthic-pelagic axis (but showed some weak morph differences). While shape variation in them could reflect biomechanical differences due to size [81,82], they did not vary along a morph size gradient. In salmonids, the function of the supramaxilla (if any) is unknown. The morph differences in supramaxilla may reflect correlations with linked bones (like the maxilla), as the high level of integration between these bones indicates. In contrast, the quadrate is a critical linkage in the suspensorium, its main function to allow rotation of the lower jaw. Since changes to the maximum angle of opening potentially needed to capture different prey would likely only require subtle morphological changes in this bone, it is unlikely we could detect these differences given our methods, or divergent selection may not have acted strongly on quadrate shape.

While adult morphology should reflect long-term prey consumption (e.g., higher pelagic prey consumption associates with stronger pelagic features), variation in the mechanisms of capturing and processing of different prey items within and between sympatric charr morphs, and potential impacts on morphological variation, is poorly understood. To our knowledge biomechanical studies of salmonids have not yet been done on fishes that specialize on such divergent prey as snails and zooplankton. Feeding was compared in PL- and SB-charr, revealing reduced efficiency of zooplankton feeding (pelagic) in SB-charr juveniles due to lower attack rates [40], increased handling time [83], increased ejection of prey [83] and poorer retention in the branchial apparatus likely due to shorter gill rakers [40]. Both morphs likely capture zooplankton using a similar strategy, i.e., suction [40], however may exhibit differences

in capture efficiency as a result of morphological differences associated with ecological divergence [40]. Prior studies are limited to biomechanics of piscivorous feeding in brook trout (*Salvelinus fontinalis*) [84] or theoretical models of suspensorium and lower jaw movement [85]. Salmonid jaw elements are likely to be largely immobile due to rigid attachments of the components [70]. Therefore, the use and efficiency of suction is likely limited compared to pelagic fishes with highly mobile jaws [86], and perhaps not highly differentiated among arctic charr morphs.

Benthic-pelagic divergence can manifests in lower jaw length relative to head length [65], as seen in subterminal mouths of benthic arctic charr [37,44,87]. Shelled freshwater snail (*Lymnaea peregra*) is the main prey of the derived SB- and LB-charr in Þingvallavatn [37,40]. In general, hypertrophied pharyngeal jaws [88–92], molariform pharyngeal teeth [93,94], and decreased gill raker length and density [40,44,95,96] are all morphological adaptations widely employed across fishes that consume large quantities of hard-shelled benthic prey. Also, mechanisms for capturing attached prey vary widely among species [63,71,72]. Use of strong suction, gripping and biting, lateral movements, and highly protrusible jaws are commonly utilized traits [63,71,72]. However, details of the intersection of these biomechanical and behavioural components for feeding on divergent prey resources in Þingvallavatn morphs is currently unexplored. In many groups of fishes, strong benthic-pelagic ecological divergence has led to modification in morphology and behaviour to allow for optimal feeding performance [16,58,59,63,65,74,76,78,97–100]. Clearly, investigation of the behavioural and biomechanical strategies individuals and morphs use to both capture and process prey, and bone shape in the same individuals, is needed to reveal how arctic charr specialize on such divergent prey types.

While we expected adaptions to divergent prey resources to be expressed in shape divergence of cranial elements among morphs, other morphological modifications can occur that do not impact bone shape as studied here. Internal bone size and architecture has been shown to diverge along the benthic-pelagic axis, with benthic feeders exhibiting more robust hollow cranial bones with complex internal structure, similar to mammalian long-bones, and planktivores having thin solid bones [101]. Bones of benthic scrapers also have higher rates of mineralization [102] and are more resistant to bending [101]. We found benthic morphs (LB- and SB-charr) to have thicker and more robust bones overall (relative to PL- and PI-charr), but variation in their internal structure and mineralization is unknown. The differences in adult bone size and shape found here may associate with differentiation in muscle mass and surface area of attachment sites, and as skeletal and musculature may contribute independently to biomechanical diversity [103], adaptive changes to soft tissues may both be more critical in foraging divergence (i.e. proportionally larger impact on biomechanics of feeding related to speed, force, and mode of prey capture) and be more apparent than associated changes in bone shape in these morphs.

## Influence of extrinsic and intrinsic factors on bone variation within and among morphs

The observed shape variation in the jaw bones within and between morphs could be influenced by differences in their ecology and food preferences [see above, 40,41], genetic composition of the morphs [36,104] and/or interactions of specific genes and environmental factors.

Adaptive plasticity may contribute to ecological and morphological divergence among sympatric and allopatric charr [21,105,106]. We studied wild caught individuals of the four morphs that mostly operate in different habitats and feed on different prey items that vary seasonally in abundance and nutrition, all factors that may influence their development and adult morphology [40,41,45]. Because the morphs differ in the location, timing and synchronicity of

spawning [43], their juveniles may receive different maternal provisions and encounter different juvenile habitats. Arctic charr shows developmental plasticity in body shape, size and behaviour in response to benthic vs. pelagic diet treatment in experimental settings [105,107,108]. Behavioural traits related to habitat choice, prey capture and processing can influence variation in bone traits [97]. For example, in sheepshead wrasse (*Archosargus probatocephalus*), individuals consuming a higher proportion of hard-shelled prey had larger adductor mandibulae and jaw bones but also employed biting mode of feeding to dislodge prey as well as buccal manipulations to crush shells and separate hard and soft parts before ingestion [109]. Thus, some of the morphological differences between (and/or within) morphs revealed here could reflect this developmental plasticity. Note however, not all environmentally induced variation in bone shape (plastic response) may be adaptive [110].

Genetic differences by groups can be assessed with experiments and population genetics. Common garden rearing of offspring of Þingvallavatn morphs indicate genetic influence on many aspects of juvenile and adult morphology [46,49,56,67,111], though maternal and/or transgenerational parental effects could also play a role [112]. Three of the morphs (PL-, SB-, and LB-charr) are genetically separable [36] and genomic data revealed substantial allele frequency differences between them, on nearly all linkage groups [35,39]. We saw the strongest morph differences in the dentary and premaxilla bones of the lower and upper jaws, respectively, that may reflect alleles influencing these bones specifically. Models indicate these three morphs have been separate for thousands of generations, with limited gene flow between them [35,36]. Curiously, the PI-charr is genetically more heterogeneous, most individuals are genetically similar to PL-charr, but others have either LB-charr or mixed PL-/LB-charr ancestry [39]. One hypothesis is that PI-charr result from ontogenetic niche shifts of juvenile PL-charr that learn to catch sticklebacks, abandoning zooplankton feeding, which may allow individuals to attain larger size and develop piscivorous characteristics [44]. Another suggestion [39] is that juvenile LB-charr and/or LB/PI hybrids also shift to piscivory. For most bones studied here, the PI-charr tended to align with PL-charr (consistent with ancestry), except in the maxilla where PI clustered closer to LB-charr. This heterogeneity of phenotypes in PI-charr could either reflect variation in genes, or the impact of piscivory on development of specific bones. Piscivory and associated size increase may place high functional demands on bone shape [113] and enhance divergence in shape and even suppress morphological variation among piscivorous charr of different genetic origins. As mapping can identify loci, genes or specific polymorphisms impacting variation in shape [114–116], QTL or association studies of bone shape variation in PI-, LB- and PL-charr could identify alleles and pathways of relevance for bone shape variation in salmonids.

Most likely, interactions between particular alleles and specific environmental factors, acting at specific times or over broader developmental windows [117], produce the observed morph differences in bone shape. Considering a hypothetical case, SB-charr may have alleles that influence the development of the dentary and articular-angular (lower jaw bones), that will induce the subterminal morphology–in individuals that eat snails as juveniles. Tests of such hypotheses require joint study of genetic and environmental influences on specific aspects of how bones develop and acquire their shape and functions. To distinguish between genetic and transgenerational environmental effects, mapping, crosses between morphs and multigenerational studies are needed. One could test the sensitivity of shape-alleles to environmental factors, like specific juvenile food types (e.g., benthic or pelagic). Analyses of gene expression can also reveal developmental pathways that respond differently to environment by morph. Already RNA-, methylation- and miRNA-sequencing of embryos of these morphs and aquaculture charr have demonstrated differences by morphs [118,119]. For instance, in PL-, LB- and SB-charr reared in a common garden ~2000 genes were differentially expressed by

morphs during development (from formation of the gill-arches to establishment of the viscero-cranium primordia) [104]. This proves the morphs differ in genes influencing multiple developmental systems that could induce these bone shape differences. Specifically, as common garden experiments revealed ossification of skull elements in juveniles starts earlier in SB-compared to PL-charr [111], it would be interesting to explore effects of genes and environmental factors on ossification and bone shape. Most probably interactions of specific environmental attributes and genetic composition cause the observed morph differences in bone shape, for instance via differences in patterns of growth or differentiation in the bone primordia or linked tissues.

## Future avenues in research of bone shape variation in salmonids

We studied variation in six dentary bones in sexually mature fish of derived morphs from one lake. This study could be expanded in several ways. First, size, age and sex can influence traits [37,120]. We studied sexually mature individuals, and average age and size varied quite substantially by morph. It would be interesting to study the emergence of bone shape variation during ontogeny in these morphs, by sampling age classes and genotyping individuals (to assign them to morph). For bone shape, size (fork length) had stronger effect on shape than age. Also, despite significant difference among morphs in allometries (age-size relationship as well as between size measures of head, body, bones), these effects were rather small, and we concluded bone size was a robust proxy for overall size (and was used to account for size variation). But with more samples it might be possible to disentangle and better estimate these effects. The same applies to potential sexual dimorphism or morph specific sex-dimorphism in bone size or shape allometry. Secondly, as four derived morphs were studied, the phenotypes could not be polarized with respect to an outgroup. Studies of anadromous charr could identify ancestral vs. derived character states. Arctic charr in Iceland have repeatedly evolved along a benthic-pelagic axis and subsequently experienced modification to allow for capture and processing of diverse prey items (e.g., benthic macroinvertebrates, fish, snails) [31,32,35,87]. It would be interesting to study the degree of parallelism in bone shape variation and integration in more sympatric benthic-pelagic morphs or the miniaturized SB-charr. Thirdly, the salmonid head contains several dozens of bones and the cranial case. Ingimarsson [47] found a signal (but weak) of morph differences in linear measures of several other bones, but geometric morphometrics on these and other bones (in 2D or 3D from CT-scans) may reveal this more systemically and highlight potentially adaptive changes [54,121,122]. Analyses of associated variation in soft tissues (muscles and ligaments), cartilage and ossification would also be interesting. We found rather high integration in shape and size variation among bones within and between the upper and lower jaws. However, covariation of shape and univariate traits (like teeth numbers) or variation in cranial kinesis, how the bones lie together in the skull and how they move together [84], was not analysed here. The feeding structures of fishes are highly functionally integrated [3,86]. Study on the seven dolly varden trout (*Salvelinus malma*) morphs in Lake Kronotskoe indicated morph specific combinations of morphological traits and differences in benthic-pelagic ecologies manifest in mouth position and jaw length divergence early in ontogeny [24]. Three cranial bone sets had with coordinated shifts in ossification: jaws (premaxilla, maxilla and dentary), intraoral tongue-bite apparatus (vomer and lingual bone, gill arch elements) and skull dermal bones (supraethmoid, frontal and preopercle), most likely reflecting differences in early ossification [24]. Studies on additional internal skull elements and their integration are needed to understand the functional capacity of different jaw elements and how they relate to the remarkable feeding divergence in arctic charr.

## Conclusions

The diversity of feeding morphology in fishes provides opportunities to examine variation in functional elements, to determine rates of evolution, integration among morphological units, and how specialization occurs along a benthic-pelagic axis. We gathered unique data on intraspecific variation in feeding related bones. The results indicate a clear benthic-pelagic separation that was most prominent in the dentary bone. Further work is needed to understand the causes of variation in feeding elements within species, how it evolves and develops. Are the axes of jaw bone shape variation in salmonids shared with other fishes or even other vertebrates? Does rapid evolution of these traits in other salmonids also lead to changes in allometric relationships, like we observed for the dentary? Work along these lines may aid in our understanding of the processes that still generate the great biodiversity we observe and must protect.

## Supporting information

**S1 Appendix. Sampling scheme and summary statistics for the four sympatric charr morphs.** Numbers of individuals, average fork length (FL), weight and age and the standard deviation (SD) for length, weight and age of the 240 individuals. [1]Fork length: Length from tip of snout to the posterior tip of middle caudal fin ray. Log: Natural logarithmic transformation. (CSV)

**S2 Appendix. Explanation of 37 landmarks used for landmarking the external shape, X-S indicates sliding landmark.** (CSV)

**S3 Appendix. Explanation of landmarks used to capture the shape of the six bones (dentary, articular-angular, quadrate, premaxilla, maxilla and supramaxilla).** X-S indicates sliding landmark. See Fig 1D-1E for placement of landmarks on bones. (CSV)

**S4 Appendix. Variation in size and age of sexually mature fish of the four sympatric morphs.** (A) Histogram, showing age (years) distribution for all morphs by sex (NA indicates SB that could not be sexed). (B) The variation in fork length (FL, cm) by morph by sex represented by a boxplot. (C) The variation in $\log_e$ weight (g) by morph by sex are represented in boxplots. (PDF)

**S5 Appendix. ANOVA results from tests of the influence of morph and sex effects on length (cm FL), $\log_e$ weight (g) and age (years).** F = F-statistic, P = P—value, $< 0.05$ in bold. (CSV)

**S6 Appendix. Results from the Procrustes ANOVA, for effects of morph and $\log_e$ FL on the size (centroid size) of the six bones.** * Despite the p-values, all $R^2$ for the interaction term were small and no pairwise test of slope by morph were significant. $R^2$ = coefficient of determination; F = F-statistic; Z = Z-statistic; P = P-value, $< 0.008$ in bold. (CSV)

**S7 Appendix. Size corrected PC-plots of specimens and deformation grids showing shape variation in (top) external whole-body shape and (bottom) head shape.** Plots of shape warps on X- and Y-axis are unmagnified. Each dot represents an individual and the ellipses represent 95% CI for the distribution by morph (large dot represents the mean of each morph distribution in these two dimensions of shape). For the whole-body (top) PC2 and 3 explain

18% and 8% of the variation respectively (PC1 was biased by sampling error, and not depicted) and for the head shape (bottom) PC3 and 4 explain 40% and 22% respectively (PC1 and PC2 were biased by sampling error, and not depicted).
(PDF)

**S8 Appendix. Shape variation in the two external shapes and 6 head bones.** Analysed with Procrustes ANOVA, testing for influence of sex and size (centroid size). $R^2$ = coefficient of determination; F = F-statistic; Z = Z-statistic; P = P-value, $\leq 0.008$ in bold.
(CSV)

**S9 Appendix. PC-plots of specimens showing shape differences between replicates (trials), for dentary, premaxilla, articular-angular, maxilla, quadrate and supramaxilla (from top to bottom, left to right).** Each dot represents an individual and the ellipses represent 95% CI for the distribution by replicates (large dot represents the mean for each morph replicates in these dimensions).
(PDF)

**S10 Appendix. Ordination plot (from partial least squares analysis (PLS)) showing the degree of association between the two replicates (trials) for dentary, premaxilla, articular-angular, maxilla, quadrate and supramaxilla (from top to bottom, left to right).** Block 1 (x-axis) is replicate 1 and Block 2 (y-axis) is replicate 2. For all bones association between the replicates was always significant (p < 0.001). With the correlation coefficient being, dentary: 0.989, premaxilla: 0.939, articular-angular: 0.994, maxilla: 0.986, quadrate: 0.942 and supramaxilla: 0.835.
(PDF)

**S11 Appendix. ANOVA results from tests of the influence of individual variation and measurement error (replicates) effects on shape variation in 6 jaw bones.** Analysed with Procrustes ANCOVA. $R^2$ = coefficient of determination; F = F-statistic; Z = Z-statistic; P = P-value, < 0.008 in bold.
(CSV)

**S12 Appendix. Size and shape allometry of (top) quadrate and (bottom) supramaxilla).** Left panels: Relationships between bone size and fork length by morph; Right panels: Relationships between bone shape and bone size. On the right are inset the associated shape changes related to each component, grey outlines the mean shape, and black the extremes for each PC. Shown are values for individuals (open circles) and the predicted values (filled) of regressions for both bone size ($CS_{bone}$) vs body size ($\log_e$ FL) or bone-shape vs bone-size. $\text{Log}_e$, natural log transformation.
(PDF)

**S13 Appendix. Estimates of regression for the relationships between the size of the individual and the size of the bone, per morph.** See also Fig 2 and S6 and S12 Appendices.
(CSV)

**S14 Appendix. Results from pairwise comparisons of variation in bone size between morphs for the six bones (dentary, articular-angular, quadrate, premaxilla, maxilla and supramaxilla).** Based on a model assuming no difference in allometry by morphs. D = pairwise distances between means; UCL = upper confidence level; Z = Z-statistic; P-value < 0.008 in bold (Bonferroni).
(CSV)

**S15 Appendix. Information on PCA, for all bone shapes, when size effects have not been removed.** Symbols indicate how morphs align, # for when morphs align along morphotype, * when morphs align along the size- gradient. Double Symbols (* or #) for prominent separation and symbol in brackets for minor.
(CSV)

**S16 Appendix. Pairwise distances from tests of dentary shape allometry by morph, between vector angles and absolute differences between vector lengths.** Following ANCOVA model with the interaction of size (bone centroid size) and morph. $r$ = slope vector correlations; *angle*: Between the two vectors being compared; $d$ = pairwise distances between means; UCL = upper confidence level; Z = Z-statistic; P-value, $\leq 0.008$ in bold (Bonferroni correction).
(CSV)

**S17 Appendix. Results from pairwise tests of shape by morphs, for differences between absolute vector lengths (D) used to examine possible shape differences by morph in five bones.** These bones had significant shape differences according to ANOVA model with main effects and interaction of morph by bone size. Used to verify results from ANOVA and test for true allometry (i.e., different group slopes). D = pairwise differences between vector lengths; UCL = upper confidence level; Z = Z-statistic; P-value, $< 0.008$ in bold.
(CSV)

**S18 Appendix. Results from "Significant Vector Angle test" examining possibly allometric shape differences by morph for five bones, that showed significant allometric differences according to ANOVA model with main and interaction effects (morph x bone size).** Used to verify results from ANOVA and test for true allometry (i.e., different group slopes). R = slope of vector correlation; UCL = upper confidence level; Z = Z-statistic; P-value $< 0.008$ in bold.
(CSV)

**S19 Appendix. Size corrected PC-plots of specimens and deformation grids showing shape variation in (top) quadrate and (bottom) supramaxilla.** On each axis are the associated shape changes related to each component, grey outlines the mean shape and black the extremes for each PC. Plots of shape warps on X- and Y-axis are unmagnified. Each dot represents an individual and the ellipses represent 95% CI for the distribution by morph (large dot represents the mean of each morph distribution in these two dimensions of shape). For the quadrate (top) PC1 and 2 explain 30% and 14% of the variation respectively and the supramaxilla (bottom) PC1 and 2 explain 40% and 22% respectively.
(PDF)

**S20 Appendix. Tests of integration among craniofacial bones in arctic charr.** The six bones were compared using partial least squares (PLS), within anatomical regions (upper and lower jaws) and between regions. *Effect size of maxilla-supramaxilla were significantly higher than premaxilla-maxilla. **Effect size of dentary-maxilla and articular-angular were significantly higher than articular-angular-supramaxilla.
(CSV)

## Acknowledgments

We thank Haraldur R. Ingvason and Stefán Már Stefánsson of the now closed Kópavogur Nature Center, and Fia Finn, Zophonías O. Jónsson and Grégoire Gaillet at the University of

Iceland for help with sampling. We thank Benóný Jónsson and coworkers at the Marine & Freshwater Research Institute of Iceland for providing fish and data. Members of the Arctic charr-group at the University of Iceland deserve our gratitude for help during the project, discussions, constructive criticism and great social environment. Finally, we thank the editor, an anonymous reviewer and Michelle C. Gilbert for helpful comments on the manuscript.

## Author Contributions

**Conceptualization:** Guðbjörg Ósk Jónsdóttir, Finnur Ingimarsson, Sigurður Sveinn Snorrason, Arnar Pálsson, Sarah Elizabeth Steele.

**Data curation:** Guðbjörg Ósk Jónsdóttir, Laura-Marie von Elm, Finnur Ingimarsson, Samuel Tersigni, Sigurður Sveinn Snorrason, Arnar Pálsson.

**Formal analysis:** Guðbjörg Ósk Jónsdóttir, Arnar Pálsson, Sarah Elizabeth Steele.

**Funding acquisition:** Arnar Pálsson.

**Investigation:** Guðbjörg Ósk Jónsdóttir, Sarah Elizabeth Steele.

**Methodology:** Guðbjörg Ósk Jónsdóttir, Laura-Marie von Elm, Finnur Ingimarsson, Samuel Tersigni, Sigurður Sveinn Snorrason, Arnar Pálsson, Sarah Elizabeth Steele.

**Project administration:** Arnar Pálsson.

**Supervision:** Guðbjörg Ósk Jónsdóttir, Arnar Pálsson, Sarah Elizabeth Steele.

**Visualization:** Guðbjörg Ósk Jónsdóttir, Sarah Elizabeth Steele.

**Writing – original draft:** Guðbjörg Ósk Jónsdóttir, Arnar Pálsson, Sarah Elizabeth Steele.

**Writing – review & editing:** Guðbjörg Ósk Jónsdóttir, Laura-Marie von Elm, Finnur Ingimarsson, Samuel Tersigni, Sigurður Sveinn Snorrason, Arnar Pálsson, Sarah Elizabeth Steele.

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
