## [Decision Letter · Decision Letter 0]

5 Jan 2024

PONE-D-23-30771Diversity in the internal functional feeding elements of sympatric morphs of Arctic charr (*Salvelinus alpinus*).PLOS ONE

Dear Dr. Jónsdóttir,

Thank you for submitting your manuscript to PLOS ONE. After careful consideration, we feel that it has merit but does not fully meet PLOS ONE’s publication criteria as it currently stands. Therefore, we invite you to submit a revised version of the manuscript that addresses the points raised during the review process.

We look forward to receiving your revised manuscript.

Kind regards,

Michael Schubert

Academic Editor

PLOS ONE

3. To comply with PLOS ONE submissions requirements, in your Methods section, please provide additional information regarding the experiments involving animals and ensure you have included details on (1) methods of sacrifice, (2) methods of anesthesia and/or analgesia, and (3) efforts to alleviate suffering

"We thank the Institute of Biology at the University of Iceland for financial support."

"AP: The Institute of Biology at the University of Iceland supported the field expeditions. No grant number. (www.luvs.hi.is). The funder had no role in study design, data collection and analysis, decision to publish, or preparation of the manuscript."

Reviewers' comments:

Reviewer's Responses to Questions

**Comments to the Author**

1. Is the manuscript technically sound, and do the data support the conclusions?

Reviewer #1: Yes

Reviewer #2: Yes

2. Has the statistical analysis been performed appropriately and rigorously? 

Reviewer #1: Yes

Reviewer #2: Yes

3. Have the authors made all data underlying the findings in their manuscript fully available?

Reviewer #1: Yes

Reviewer #2: Yes

4. Is the manuscript presented in an intelligible fashion and written in standard English?

Reviewer #1: Yes

Reviewer #2: Yes

5. Review Comments to the Author

Reviewer #1: This is a nice and interesting study that I enjoyed reading. In my opinion it is basically ready for publication and I do not have any major comments. I do have one rather minor comment, though: You mention that the patterns of allometry differ among morphs (and bones). You touch this topic only marginally in the discussion (or did I miss it?), although I think this is very intersting. I would suggest to put also more emphasis on discussing the differences in allometry - i.e. by discussing potential (obvious?) niche shifts and corresponding morphological changes during ontogeny.

Reviewer #2: The authors approach their questions, and the data, appropriately and are quite thorough in their methodology.

Statistical analyses were appropriate, and the authors used well established methods for their data type and hypothesis testing.

The authors provide all required information on how to find their data, images, and methods.

The manuscript is extremely well written.

6. PLOS authors have the option to publish the peer review history of their article (what does this mean?). If published, this will include your full peer review and any attached files.

Reviewer #1: No

Reviewer #2: **Yes: **Michelle C Gilbert

---

## [Author Response · Author response to Decision Letter 0]

19 Feb 2024

Here when referencing pages or lines of the manuscript, we are refereeing to track changes version of the manuscript (BoneManuscript_track_changes.docx).

We went over the entire manuscript to check for typos and other glitches and change a few sentences we felt could be worded better. All such changes are indicated in the track changes version of the manuscript.

Our responses: 

R: We have reviewed the manuscript and file names. All seem to meet the PLOS ONE’s style requirements. 

R: We have uploaded the image files and associated datasets to Figshare+ (doi: 10.25452/figshare.plus.25118825) and flat files with landmark data and other data, and the R code to Github (https://github.com/GudbjorgOskJ/CharrBonesTVV).

3. To comply with PLOS ONE submissions requirements, in your Methods section, please provide additional information regarding the experiments involving animals and ensure you have included details on (1) methods of sacrifice, (2) methods of anesthesia and/or analgesia, and (3) efforts to alleviate suffering.

R: We amended the methods section (page: 5, line: 105, track changes version of the manuscript) It now reads:

“Sexually mature arctic charr from Lake Þingvallavatn were collected (Fall 2020/21) using composite nets laid overnight (fish died in net) and seine nets (fish were killed by single blow to the head).”

4. Thank you for stating the following in the Acknowledgments Section of your manuscript: "We thank the Institute of Biology at the University of Iceland for financial support." We note that you have provided additional information within the Acknowledgements Section that is not currently declared in your Funding Statement. Please note that funding information should not appear in the Acknowledgments section or other areas of your manuscript. We will only publish funding information present in the Funding Statement section of the online submission form. Please remove any funding-related text from the manuscript and let us know how you would like to update your Funding Statement. Currently, your Funding Statement reads as follows: "AP: The Institute of Biology at the University of Iceland supported the field expeditions. No grant number. (www.luvs.hi.is). The funder had no role in study design, data collection and analysis, decision to publish, or preparation of the manuscript." Please include your amended statements within your cover letter; we will change the online submission form on your behalf.

R: We removed the “We thank the Institute of Biology at the University of Iceland for financial support." from the Acknowledgements. There is no need to update the Funding statement. 

Note, we now thank editor and reviewers in Acknowledgements. Updated Acknowledgements (page: 24, line: 547, track changes version of the manuscript):

“…. constructive criticism and great social environment. Finally, we thank the editor, an anonymous reviewer and Michelle C. Gilbert for helpful comments on the manuscript.”

R: The ethics statement was slightly rewritten, expanded and text clarified, and is now also included in the Methods section (page: 5, line: 102, track changes version of the manuscript): 

“This study involves sampling and killing of wild fishes, and for such studies a scientific fish fieldwork from the Directorate of Fisheries in Iceland (www.fiskistofa.is) is needed. Arnar Pálsson, an author on this study was in charge and present for all sampling efforts and, for the period of study, had such scientific fish fieldwork permits (#0460/2021-2.0 and #0042/2014-2.13).”

6. When completing the data availability statement of the submission form, you indicated that you will make your data available on acceptance. 

R: We have made the data accessible on Figshare+ (doi:10.25452/figshare.plus.25118825) and on Github (https://github.com/GudbjorgOskJ/CharrBonesTVV). See page: 6, line: 124 in the track changes version of the manuscript. 

7. Please review your reference list to ensure that it is complete and correct.

R: We have reviewed the reference list and made a few minor changes. One reference (Pigliucci, 2008) was removed as we felt the citation was not necessary for that sentence, used to be cited at page: 2, line: 35. One new citation was added (reference number 108, Delling et al., 2023), cited on page: 20, line: 452.

We edited references number 89 (Huckins, 1997) and 113 (Collar et al., 2009). In the previous version all characters in the title of the papers were uppercase, this has now been fixed.

Reviewer #1: This is a nice and interesting study that I enjoyed reading. In my opinion it is basically ready for publication and I do not have any major comments. I do have one rather minor comment, though: You mention that the patterns of allometry differ among morphs (and bones). You touch this topic only marginally in the discussion (or did I miss it?), although I think this is very intersting. I would suggest to put also more emphasis on discussing the differences in allometry - i.e. by discussing potential (obvious?) niche shifts and corresponding morphological changes during ontogeny.

R: We thank the reviewer for this good summary. We like the suggestion of elaborating on the allometry, and have adjusted the manuscript to expand slightly on this angle. See page: 17, line: 376 in the track changes version of the manuscript:

“It would be interesting to see if this holds in more salmonid populations, and also if this is accompanied by notable allometric changes in the dentary as we found here.”

And (page: 23, line: 537, in the track changes version of the manuscript):

“Does rapid evolution of these traits in other salmonids also lead to changes in allometric relationships, like we observed for the dentary?”

Reviewer #2: The authors approach their questions, and the data, appropriately and are quite thorough in their methodology. Statistical analyses were appropriate, and the authors used well established methods for their data type and hypothesis testing. The authors provide all required information on how to find their data, images, and methods. The manuscript is extremely well written.

R: We thank the reviewer for this generous and warm summary.

---

## [Editor Report · Decision Letter 1]

27 Feb 2024

Diversity in the internal functional feeding elements of sympatric morphs of Arctic charr (*Salvelinus alpinus*).

PONE-D-23-30771R1

Dear Dr. Jónsdóttir,

We’re pleased to inform you that your manuscript has been judged scientifically suitable for publication and will be formally accepted for publication once it meets all outstanding technical requirements.

Kind regards,

Michael Schubert

Academic Editor

PLOS ONE

---

## [Editor Report · Acceptance letter]

9 May 2024

PONE-D-23-30771R1 

PLOS ONE

Dear Dr. Jónsdóttir, 

I'm pleased to inform you that your manuscript has been deemed suitable for publication in PLOS ONE. Congratulations! Your manuscript is now being handed over to our production team.

Kind regards, 

on behalf of

Dr. Michael Schubert 

Academic Editor

PLOS ONE